# The Application of Microfibrous Entrapped Activated Carbon Composite Material for the Sarin Simulant Dimethyl Methylphosphonate Adsorption

**DOI:** 10.3390/nano13192661

**Published:** 2023-09-28

**Authors:** Yucong Xie, Chao Zheng, Liang Lan, Hua Song, Jian Kang, Kai Kang, Shupei Bai

**Affiliations:** 1School of Light Industry and Engineering, South China University of Technology, Guangzhou 510640, China; ycong6364@163.com (Y.X.); 13263157913@163.com (C.Z.); 17307404197@163.com (L.L.); 2State Key Laboratory of NBC Protection for Civilian, Beijing 102205, Chinakangjian@sklnbcpc.cn (J.K.)

**Keywords:** microfibrous composite, Dimethyl methylphosphonate, structured fixed bed, adsorption equilibrium, adsorption kinetics

## Abstract

Granular activated carbon (GAC) has proven to be an effective adsorbent for removing the chemical warfare agent sarin (GB) and simulants like Dimethyl methylphosphonate (DMMP). However, it comes with certain limitations, including inadequate contact efficiency, notable mass transfer resistance, and lower bed utilization efficiency. This study synthesized steel fiber-entrapped activated carbon composites (SFEACs), which exhibited a maximum adsorption capacity of 285.3 mg/g at 303 K. Compared with the packed bed (PB) filled with GAC, while the adsorption capacity of SFEACS decreased, there was a substantial increase in the adsorption mass transfer rate. These SFEACs were combined with GAC to create a structural fixed bed (SFB), which demonstrated excellent performance in DMMP removal. Under identical experimental conditions, the DMMP breakthrough curve of SFB exhibited a steeper profile compared to the packed bed (PB) filled with GAC at the same bed height, and the breakthrough time against DMMP vapor could be extended by 13.8%. Furthermore, the adsorption rate constant of the Yoon-Nelson model increased by more than 17.6%, and the unused bed length, according to the Wheeler–Jonas model, decreased by more than 14%.

## 1. Introduction

In recent years, chemical warfare agents (CWAs) have been used in conflicts and terrorist attacks, posing a significant threat to human life. Among them, nerve agents made from organophosphorus compounds such as Sarin(GB), known for their high and rapid toxicity, have become one of the most dangerous CWAs [1,2]. Due to this, GB is unsuitable for direct use in scientific research [3]. DMMP was commonly regarded as a more suitable simulant for GB, especially under conditions where only physical adsorption was taken into account, without considering catalytic decomposition [4,5,6,7,8]. There are currently many techniques for getting rid of CWAs, including the hydroponic process [9], plasma technique [10], ozone oxidation [11], Laser-induced heating [12], photocatalytic degradation [13], and catalytic/adsorption purification methods [14,15,16,17]. However, adsorption purification technology that can swiftly adsorb hazardous compounds in various situations is thought to be one of the most effective options when considering aspects like cost, energy consumption, and practical applicability [18].

Packed beds (PB) filled with granular activated carbon (GAC) as mature adsorption and purification units have been widely used in areas such as chemical protective equipment and treatment of industrial wastes, etc. [19,20,21]. However, the unit demonstrates insufficient mass transfer efficiency, higher resistance to mass transfer, and a lower rate of bed utilization; therefore, improving contact efficiency is crucial for enhancing the adsorption transfer process [22,23,24]. The use of microfibrous entrapped catalysts/sorbents (MFECS), developed by Tatarchuk at Auburn University in the United States, shows promising potential for applications in this field [25,26,27]. The utilization of a fiber network structure composed of microfibers for entrapping micron-sized multiphase catalysts, adsorbents, and other solid reactant particles results in a significant improvement in heat and mass transfer processes. Compared with traditional adsorbents, MFECs have a significant advantage of high adsorption efficiency and low diffusion resistance [28,29,30]. In order to effectively increase adsorption performance, bed utilization, and protection time, researchers have combined the MFECS and PB to create an SFB, where the PB layer removes the high-concentration of adsorbate at the front end of the bed, and the low-concentration of adsorbate is removed at the tail end [31,32,33]. For instance, Yan et al. [34] investigated the toluene adsorption effectiveness of MFECS and PB beds made up of structured beds. Compared to the PB to the particle bed at the same bed height, the unused bed length of the SFB was lowered by almost 20%. 

Although there have been many scientific studies on the application of SFB to volatile organic compounds adsorption and catalytic processes [35,36,37], existing literature indicates that there are few reports on the adsorption of CWAs on SFB [31]. Therefore, in this study, stainless steel fiber-entrapped activated carbon composites (SFEACs) were prepared, and equilibrium and kinetic analysis was carried out to investigate the adsorption mechanism of DMMP vapor by SFEACs. Meanwhile, SFB was constructed. In order to investigate the enhanced mechanism of the DMMP vapor adsorption process by SFB, the kinetic analysis of DMMP breakthrough curves on PB and SFB under various conditions was performed using dynamic models, and various parameters affecting the adsorption process were examined, such as inlet concentration, flow rate, and adsorption temperature. This study offers valuable support for the advancement of more efficient chemical protection technology.

## 2. Materials and Methods

### 2.1. Materials

Columnar granular activated carbon was purchased from Xinhua Chemical Factory (Taiyuan, China). Stainless steel fibers and plant fibers with various diameters were obtained from Hunan Huitong Advanced Materials Co., Ltd. (Changsha, China). Dimethyl methyl phosphate of analytical purity was acquired from Meryer Reagents (Shanghai, China).

### 2.2. SFEACs Preparation

The GAC with an average particle diameter of 0.9 mm was milled and sieved to fine particles of 125–150 μm as adsorbent. The detailed synthesis process has been reported in our previous paper involving a wet layup-high temperature sintering process [38]. SFEACs consist of GAC, wood fibers, and stainless steel fibers. The wood fibers were used as the binder of the wet layup precursor and were removed after the sintering process.

### 2.3. Characterization 

An Autosorb iQ instrument (Quantachrome Instrument Corporation, USA) was utilized for the N_2_ adsorption/desorption isotherm analysis at 77 K, which served to ascertain the pore structure characteristics of the SFEACs. The specific surface area was computed using the BET method, and the DFT model was utilized to determine the pore volume and pore size distribution. The composite morphologies were inspected using a Zeiss Merlin Compact scanning electron microscope (SEM). A Perkin Elmer Frontier spectrometer was used for Fourier transforms infrared spectroscopy (FTIR) to study the surface chemistry of the samples. The thermal characteristics of the SFEACs were assessed with a TG/DTA instrument (Thermo plus EV2). Roughly 40 mg of the samples were measured and positioned in ceramic crucibles for these evaluations. The analysis was conducted over a temperature range from 30 to 1000 °C, at a heating rate of 10 °C/min under a helium environment.

### 2.4. Adsorption Experiments Setup

The experimental setup for this study is shown in Figure 1. The air is dried using the dryer and divided into two paths A,B. Path A passes through the DMMP vapor saturator three times to produce DMMP saturated vapor [39], and path B is only a high-flow air line, after which the paths A,B together enter a mixing cell ca. 400 cubic centimeters, the red pipeline is the heating pipeline with a temperature higher than vapor generation device to avoid DMMP condensation. The mixed DMMP vapor flows into an adsorption column with an inner diameter of 2 cm and a height of 20 cm, and the temperature of the adsorption column is controlled using an isothermal heating module. A total hydrocarbon analyzer (MODEL 3-600, JUM) was used to measure the concentration of DMMP vapor at the inlet and outlet produced during the experiment, and the results were recorded on a computer. Multi-stage adsorption purification was used to clean the tail gas ped into the atmosphere. The SFB was filled with 3 cm GAC (particle size 0.9 mm) and 0.5 cm SFEAC at the inlet and outlet of the adsorption column, respectively, while the PB was only filled with 3.5 cm GAC (0.9 mm).

### 2.5. Breakthrough Tests

If not otherwise mentioned, the dynamic adsorption device described in Section 2.4 was employed for the sorption experiments at various conditions. Prior to each experiment, the sorbent materials were heated with flowing N_2_ (150 mL/min) to 150 °C for 2 h, and the impurity gas was removed. The breakthrough tests were repeated at least three times for each sample; thus, the experimental points reported are generated by taking the mean value. The adsorption equilibrium of the adsorbent is considered to be reached when the DMMP vapor concentration is equal at the inlet and outlet of the bed. 

### 2.6. Adsorption Dynamics

In this part, the effects of three experimental variables—inlet concentration, flow rate, and adsorption temperature—on the dynamic adsorption process of DMMP were examined. The flow rate varied at 1.88, 2.83, and 3.77 L/min. The adsorption temperature varied from 303, 313, and 323 K. The inlet concentration is 3.6, 5.1, and 7.1 mg/L, respectively. The experimental results are expressed in the form of variations of *C*/*C*_0_ versus time. The breakthrough concentration of DMMP vapor was taken as 0.04 mg/m^3^.

### 2.7. Adsorption Isotherm

In this study, the adsorption isotherms of DMMP on SFEACs at 303, 313, and 323 *K* were measured using breakthrough tests. The mass of SFEACs filled in the adsorption bed was 4.86 g, and the equilibrium adsorption capacity of DMMP on SFEACs at six different concentrations of each adsorption temperature was measured, respectively. The adsorption capacity (*q*_0*,exp*_) of DMMP is calculated using breakthrough by Equation (1).
(1)$$q_{0,exp}=\frac{Q}{M}\int_{0}^{t_{s}}{(C}_{0}{\;-\;C}_{t})\mathrm{dt}$$

*Q*: the DMMP vapor flow rate, L/min; *t*: adsorption time, min; *t_s_*: the time of adsorption equilibrium, min; *C_t_*: the DMMP concentration at time *t*, mg/L; *M*: the mass of adsorbent, g.

## 3. Theory

### 3.1. Adsorption Kinetic Models

The adsorption behavior of gases in dynamic adsorption columns has been predicted using various kinetic models based on various hypotheses, both to correctly predict the mass transfer behavior within the dynamic adsorption column and to be used to assess how each variable affects adsorption. This research focused on examining the adsorption of DMMP vapor in a fixed bed column and the pattern of breakthrough curves, using the Yoon–Nelson and the Wheeler–Jonas models as descriptive tools.

Yoon–Nelson model [40]. This model presupposes the possibility of a breakthrough curve and adsorbate adsorption. The bed’s sorbent, adsorbent type, and physical characteristics do not need to be detailed for the Yoon-Nelson model. Equation (2) represents the Yoon-Nelson model’s non-linear form as follows:(2)$$\frac{C_{t}}{C_{0}}=\frac{1}{{1+e}^{k^{\prime }(\tau -t)}}$$

*C*_0_ is inlet concentration, g/cm^3^; *C_t_* is outlet DMMP concentration, mg/m^3^; *k′* is the Yoon-Nelson rate constant, min^−1^; *τ* is the time required for 50% adsorbate breakthrough, min.

Wheeler–Jonas model. The Wheelers-Jonas (also known as reaction kinetics) model is often used to estimate the penetration time of organic vapors in fixed beds of activated carbon particles, and the use of this model has now been extended to other types of adsorbents. The equation has now been extended to apply to other vapors and other environmental situations to estimate the service life of the bed against organic vapor [41,42].
(3)$$t_{b}=\frac{M\cdot W_{e}}{Q\cdot c_{0}}-\frac{W_{e}\cdot \rho_{b}}{k_{v}c_{0}}ln\left(\frac{c_{0}-c_{t}}{c_{t}}\right)$$

The length of the unused bed (*L_c_*) of the sorbent bed can also be easily calculated using Wheeler’s equation when the breakthrough concentration (*C_b_*) of the bed is constant.
(4)$$L_{c}=\left(v*\frac{1}{k_{v}}\right)\left[ln\left(\frac{c_{0}-c_{b}}{c_{t}}\right)\right]$$

*C*_0_ is inlet concentration, g/cm^3^; *C_t_* is outlet DMMP concentration, mg/m^3^; *t_b_* is breakthrough time, min; *M* is the amount of adsorbent, g; *k_v_* is adsorption rate coefficient, min^–1^; *W_e_* is the equilibrium adsorption capacity, mg/g; *Q* is the volumetric flow rate, mL/min; *ρ_b_* is bulk density of the adsorbent, g/cm^3^; *C_b_* is the DMMP breakthrough concentration, mg/m^3^; *v* is the DMMP vapor surface velocity, cm/min.

### 3.2. Adsorption Isotherms Models

Adsorption isotherm is generally highly significant in the design of adsorption systems. In this study, the adsorption of DMMP vapor by SFEACs was measured at three adsorption temperatures of 303, 313, and 323 K (six concentrations were selected for each adsorption temperature) using the adsorption setup in Section 2.4. Each dynamic adsorption test was performed at least three times to verify an experiment’s correctness.

Langmuir model. An isothermal model for adsorption that has been widely used is the Langmuir model [43]. This presumes that the adsorption sites are evenly distributed across the adsorbent surface and that only single molecule layer adsorption takes place on the adsorbent surface. This model’s non-linear form is as follows:(5)$$q_{e}=\frac{q_{m}K_{L}C_{e}}{{1+K}_{L}C_{e}}$$
where *q_e_* is the capacity for equilibrium adsorption; *q_m_* is the adsorbent’s ability to adsorb monolayers, mg·g^−1^; *K_L_* is the adsorption constant in Langmuir, L·mg^−1^; *C_e_* is the DMMP concentrations at equilibrium in the gas phase.

Freundlich model. The Freundlich model [44,45] is an isothermal, semi-empirical adsorption model. Freundlich suggested the model in 1906. This model accounts for the solid surface’s true non-uniformity. The Freundlich model presupposes that adsorbent is deposited as a multi-molecular layer on the surface and that the heat of adsorption is inversely proportional to the extent of coverage. This expression’s equation is as follows: (6)$$q_{e}{=K}_{F}C_{e}^{1/n}$$

*K_F_* and *n* are the Freundlich constants characteristic of the system, *K_F_* and *n* are indicators of adsorption capacity and adsorption intensity, respectively.

Dubinin-Radushkevich model. The Dubinin-Radushkevich (D-R) isotherm model has good applicability in calculating the adsorption capacity of micropores and determining the structural characteristics of adsorbent micropores. The D-R model assumes that adsorption is a physical process, and its applicability is usually limited to microporefilling adsorption processes at low coverages. In micropores with micropore diameters close to the diameter of adsorbed molecules, the potential fields of the two opposing pore walls acting on the adsorbed molecules overlap, resulting in large adsorption energy of gas molecules. This expression’s equation is as follows:(7)q=q0exp−AE2
(8)$$A=R\cdot T\cdot ln\left(\frac{p_{0}}{p}\right)$$
where *q* signifies the quantity of adsorption, mg·g^−1^; *q*_0_ stands for the maximum adsorption capacity, mg·g^−1^; *E* corresponds to the characteristic energy, which gauges the intensity of adsorption between the adsorbate and adsorbent, denoted in J/mol; *p* is used to define the absolute pressure, kPa; *p*_0_ stands for the saturated vapor pressure, kPa; *T* is used for indicating the system’s temperature, Kelvin (K); *R* signifies the gas constant, which is 8.314 J/(mol·K).

According to the adsorption data of DMMP in the measured temperature range on SFEACs, the adsorption characteristic curve is obtained using the following equation:(9)$$ln\left(C_{u}\right)=ln\left(C_{us}\right)+\left(-\frac{{\left(R_{g}\right)}^{2\;}}{{\left(\beta E_{0}\right)}^{2}}\right){\left(T\;ln\frac{P}{P_{0}}\right)}^{2}$$
where *C_u_* is the adsorption amount of the adsorbent, mg/g; *C_us_* is the maximum saturated adsorption capacity of the adsorbent, mg/g. *P* is the absolute pressure of DMMP, Pa; *P*_0_ is the saturated vapor pressure of DMMP at the experimental temperature, Pa.

### 3.3. Thermodynamics

Understanding the adsorption mechanism is crucial in the field of adsorption studies. The adsorption mechanism can be elucidated by employing thermodynamic parameters, including the Gibbs free energy change (Δ*G*°), enthalpy change (Δ*H*°), and entropy change (Δ*S*°), as depicted in Equations (10) and (11) and can be used to derive the thermodynamic parameters from the Langmuir isotherms, which incorporate experimental data and are the best isotherm model fitted [46,47,48,49]:(10)$$\Delta G^{{}^{\circ}}=-RT\ln K_{e}$$
(11)$$\Delta G^{{}^{\circ}}=\Delta H^{{}^{\circ}}-T\Delta S^{{}^{\circ}}$$

*T* denotes the absolute temperature, measured in Kelvin, and *R* is the universal gas constant, with a value of 8.314 J mol^−1^·K^−1^. The Δ*G°* (kJ mol^−1^) is the direct calculation of the parameter from the equation.

Equation (12) is used in the following location within Equation (11) to get the well-known va, not Hoff equation: (12)$$lnK_{e}=-\frac{\Delta H^{{}^{\circ}}}{R}\cdot \frac{1}{T}+\frac{\Delta S^{{}^{\circ}}}{R}$$

The parameters, Δ*H*° (kJ mol^−1^) and Δ*S*° (kJ mol^−1^ K^−1^), are determined by plotting the slope and intercept of, $\;\ln K_{e}$ against 1/*T*, respectively.

*K_e_* is the dimensionless thermodynamic equilibrium constant; it can be calculated by the following Equation (13).
(13)$$K_{e}=\frac{\left(K_{L}\cdot M\cdot {\left[\mathrm{Adsorbate}\right]}^{{}^{\circ}}\right)}{\gamma }$$

*K_L_* refers to the Langmuir equilibrium constant, measured in L·g^−1^. *γ* symbolizes the dimensionless activity coefficient, which is presumed to be 1 [50]. The term [Adsorbate]^0^ designates the standard concentration of the adsorbate, measured as 1 mol·L^−1^. M stands for the molecular weight of the adsorbate, expressed in g·mol^−1^.

## 4. Results and Discussion

### 4.1. Characterization of the SFEACs

The SEM images in Figure 2a,b reveal the distinctive three-dimensional network structure created by the microfibers, effectively trapping the micron-sized GAC particles. The sintering process between randomly arranged fiber rods, characterized by inter-fiber angles, results in the formation of sintering nodes at the overlaps. This formation gives rise to a highly porous three-dimensional network structure with sintered locking, providing significant porosity.

Figure 2c showcases the nitrogen adsorption isotherms for the SFEACs, demonstrating type I adsorption isotherms, which are marked by a significant increase in N_2_ adsorption at lower relative pressures, and particle size seems to have no effect on pore structure [51]. Figure 2d depicts the pore size distribution for both GAC and SFEACs. The distribution curves have peaks at 0.94, 1.17, and 1.5 nm, suggesting a high concentration of microporous structures within the 1–2 nm pore size range. The pore structure parameters outlined in Table 1 reveal that the micropore volume for the SFEACs is 0.188 cm^−3^·g^−1^, constituting about 65% of the total pore volume. Furthermore, the average pore size for the SFEACs is approximately 2.1 nm.

Figure 2e illustrates the FTIR spectra of the SFEACs. The absorption peaks observed approximately at 3400 cm^−1^ are likely due to the O-H stretching vibrations from the hydroxyl groups and N-H stretching vibrations from the nitrogen-containing functional groups [52]. The absorption band within the range of 2870–2960 cm^−1^ corresponds to C-H symmetric stretching vibrations. The absorption peaks in the range of 1030–1110 cm^−1^ are associated with C=O stretching vibrations in the GAC. Additionally, the absorption peak at 1625 cm^−1^ is indicative of C=C vibrations in aliphatic groups [34,53]. 

The TG curve of the SFEACs, as depicted in Figure 2f, exhibits a single weight-loss stage accompanied by a corresponding endothermic peak in the DTA curve. Between the temperature range of 50–120 °C, the removal of moisture from the sample structure takes place. From 30–670 °C, an exothermic and gradual weight-loss stage is observed, while the range of 670–1000 °C represents a slow weight-loss stage with an endothermic nature [54]. Throughout the entire thermal analysis experiment, the total weight loss rate amounts to 3.08%, indicating that only a minor oxidation reaction occurs in the SFEACs.

### 4.2. Adsorption Kinetics

#### 4.2.1. The Breakthrough Curves of PB and SFB

Under different adsorption conditions, the breakthrough curves of SFB and PB on DMMP vapor are shown in Figure 3. The results show that SFB has a lower DMMP adsorption capacity than PB, and the increase in the breakthrough time (*t_b_*) can reach 13.8%. This is because the particle size of activated carbon in SFEACs (125–150 μm) is seven times smaller than that of GAC in PB (0.9 mm), so the SFEACs have less particle diffusion resistance and higher adsorption efficiency. In addition, due to the high porosity of SFEACs itself, the gas-solid contact efficiency is relatively high compared with PB, and the low concentration of DMMP at the outlet of the fixed bed can be quickly adsorbed to prolong the breakthrough time. When the activated carbon adsorbed in SFEACs reaches the saturated adsorption amount, the breakthrough concentration will increase rapidly, so the breakthrough curve of SFB is steeper than that of PB, indicating a higher adsorption rate and higher adsorbent utilization.

#### 4.2.2. Yoon-Nelson Model 

In order to analyze DMMP breakthrough curves in PB and SFB, the theoretical model established by Yoon-Nelson in Equation (2) was adopted for non-linear regression analysis, as shown in Figure 4. The values of model parameters and correlation coefficient (*R*^2^) are shown in Table 2. Under the conditions of adsorption temperature of 313 K, inlet concentration of 3.6 mg/L, and flow rate of 2.83 L/min, the adsorption rate constant *k*′ of 3.5 cm PB and SFB are 0.14 and 0.31, respectively. According to Yoon and Nelson’s theory [55], *k*′ is related to the shape of the breakthrough curve, and steep breakthrough curves usually have a larger *k*′, which is consistent with experimental results.

In this study, the *R*^2^ of the two-bed breakthrough curves at different flow rates, inlet concentrations, and adsorption temperatures exceeded 0.9, and the *k*′ of SFB increased by more than 17.6% compared with PB. The value of 50% breakthrough time (*τ*) increases with the decrease in flow rate and inlet concentration. As the adsorption temperature increases, the value decreases gradually. The data in Table 2 also show that the calculated value is in good agreement with the experimental results. In addition, *k*′ increases rapidly as the inlet flow increases and is proportional to the square root of the velocity. This indicates that the rate-limiting step for adsorption is the diffusion of DMMP vapor to the adsorbent surface [56].

#### 4.2.3. Wheeler–Jonas Model

The Wheeler–Jonas model was used to analyze the initial part of the adsorption breakthrough curves of PB and SFB, and the corresponding fitting results are shown in Figure 5. The model is used to estimate key parameters such as adsorption rate constant (*k_v_*) and adsorption capacity (*W_e_*). The values of parameters and *R*^2^ are listed in Table 3. Under the experimental conditions of inlet concentration of 3.6 mg/L, adsorption temperature of 313 *K*, and flow rate of 3.77 L/min, the *k_v_* and *W_e_* of 3.5 cm PB and SFB were 8210.7 and 19641.3 min^−1^, 429.3 and 394.3 mg/g, respectively. The *t_b_* were 116 min and 132 min, respectively. The results show that under the same bed volume, the adsorption rate of SFB is significantly higher than that of PB. Although the saturated adsorption capacity decreases, the breakthrough time is obviously extended, and the utilization efficiency of the bed is improved.

In addition, the *W_e_* value of the composite bed is basically stable with the increase in flow rate. The *W_e_* value decreases with the increase in adsorption temperature, which is consistent with the results of thermodynamic analysis. The *W_e_* value increases with the increase in the inlet concentration [34].

The breakthrough curves of PB and SFB were analyzed according to Equation (4) to determine the length of the unused bed (*L_c_*), and the results are shown in Table 3. A steeper breakthrough curve corresponds to a shorter mass transfer area, and most of the adsorption capacity of the bed is utilized at the breakthrough point, which improves the utilization rate of the bed [30]. It can be seen that adding SFEACs at the fixed bed outlet can improve the bed utilization rate and prolong the bed breakthrough time. When the flow rate is 1.88, 2.83, and 3.77 L/min, the *L_c_* is 0.29, 0.31, and 0.34 cm, respectively, and increases with the increase in flow rate, which may be due to insufficient DMMP residence time at a high flow rate, resulting in a decrease in overall bed utilization and an increase in *L_c_*.

### 4.3. Adsorption Isotherm

The adsorption isotherm of DMMP onto SFEACs is illustrated in Figure 6b. On the one hand, as the adsorption temperature rises, a discernible reduction in adsorption capacity becomes evident. On the other hand, as the DMMP concentration elevates, the adsorption capacity experiences a noteworthy augmentation, eventually stabilizing at a maximum value, which belongs to the typical type I adsorption isotherm (IUPAC classification).

Figure 6c displays the fitting results of the Langmuir model for the adsorption isotherm. As shown in Table 4a, the model’s *R*^2^ values exceeded 0.964 at different temperatures; the theoretical adsorption capacity (*q_e_*_,*cal*_) was consistent with the experimentally determined adsorption capacity (*q_e_*_,*exp*_), indicating that the adsorption of DMMP on SFEACs follows a typical I-type in the temperature range of 303 *K* to 323 *K* [57]. Furthermore, the adsorption constant (*K_L_*) steadily decreases, and the adsorption affinity weakens when the temperature increases, indicating that the increase in temperature is detrimental to adsorption. The pore structure and temperature effects have a great influence on adsorption capacity (*q_e_*). However, it is worth noting that although the Langmuir model has a good fitting effect, it does not fully conform to the assumptions of the Langmuir model when applied to the actual adsorption behavior of DMMP on activated carbon. Therefore, D-R adsorption model was further used for fitting analysis.

Figure 6d presents the fitting results of adsorption isotherms at various temperatures using the D-R model. Specific parameter values are detailed in Table 4a. It’s evident that the concentration of DMMP reaches a specific threshold, and a notable phenomenon unfolds: DMMP molecules swiftly occupy the micropores of SFEACs, leading to a sharp surge in the adsorption capacity. Subsequently, the adsorption capacity levels off and eventually stabilizes at a constant value. This stabilization signifies that the available adsorption sites within the SFEACs micropores are progressively saturating, thereby placing constraints on further adsorption processes. This observed behavior aligns remarkably well with the fundamental principles of microporous filling theory, which posits that as adsorbent molecules gradually occupy the pores, the adsorption capacity will gradually approach saturation. 

The adsorption data of DMMP on SFEACs across all temperature ranges were plotted using Equation (9), generating the adsorption characteristic curves showcased in Figure 6e. With the *R*^2^ > 0.941, the adsorption characteristic curves of DMMP on SFEACs at any temperature can be well overlapped, and the adsorption characteristic curves are temperature invariant, which is consistent with Dubinin’s theory. With the adsorption characteristic curves, the prediction of DMMP adsorption at other temperatures can be achieved.

The fitting results of the Freundlich model are illustrated in Figure 6f. It can be seen from Table 4a that the value of *n* was always greater than 1, indicating that the interaction between DMMP and SFEACs showed a non-linear trend, the distribution of adsorbent molecules on the surface of the adsorbent was not uniform, the difference of affinity between adsorption sites and the interaction of adsorbent molecules had a significant impact on the shape of the adsorption isotherm. In addition, the value of *K_F_* decreases gradually with the increase in temperature, and the affinity between adsorbent molecules and adsorbent surface weakens. This is due to the enhanced thermal motion of the molecules, which may make the molecules interact less on the surface, resulting in a reduction in adsorption affinity.

### 4.4. Adsorption Thermodynamics

To ascertain the thermodynamic characteristics, DMMP adsorption thermodynamic studies were conducted at three different temperatures (303, 313, and 323 K). A direct determination of the Δ*G°* parameter was made using Equation (9). With an excellent linear correlation coefficient *R*^2^ of 0.9853, the results of the linear relationship fit between ln *K_e_* and 1/*T* are displayed in Figure 6a. The parameters Δ*H°* and Δ*S°* were derived following the interception and slope of the linear plots, respectively. The negative sign of Δ*G°* in Table 4b showed that the DMMP vapors’ suitable and spontaneous adsorption process onto SFEACs [58]. Physical adsorption was identified as the primary mechanism for the adsorption of DMMP on SFEACs, as indicated by the data. The ΔG° values falling within the range of −20 kJ·mol^−1^ to 0 kJ·mol^−1^ suggest that the adsorption process predominantly occurs via physical adsorption. The computed Δ*H°* value was −23.33 kJ·mol^−1^; the negative value denoted exothermic adsorption, indicating strong physical adsorption dominating the adsorption process. The value of Δ*S°* showed that the SFEACs’ capture of DMMP molecules reduced the system’s disorder throughout the adsorption process and the adsorbate’s randomization at the solid/gas interface [59].

## 5. Conclusions

This study delved into the adsorption mechanism of DMMP vapor within the SFEACs. To analyze the experimental data, specifically the breakthrough curves, and to determine the dynamic parameters, the Yoon-Nelson and Wheeler–Jonas models were employed. An analysis of the breakthrough curves also allowed for the determination of the length of the unused bed, offering an additional understanding of the adsorption process. The results obtained are as follows:

The DMMP vapor was adsorbed on SFEACs in the form of microporous filling, and the Langmuir and DR model was well fitted (*R*^2^ > 0.98), and the adsorption capacity of DMMP increased rapidly at lower pressures, which was typical of the type I isotherm. The thermodynamic analysis confirmed the spontaneity of the adsorption process (Δ*G*° < 0), along with its exothermic nature (Δ*H*° < 0). The adsorption was determined to be predominantly physical, as indicated by the enthalpy change falling below −20 kJ mol^−1^. 

The breakthrough curves for DMMP vapor in the SFB are more abrupt compared to those in the PB. Moreover, the length of unused bed (*L_c_*) value of SFB is reduced by more than 14%, the adsorption rate of the Yoon-Nelson model was increased by more than 17.6%, and the breakthrough time against DMMP vapor could be extended by 13.8%. The findings from both experimental and model fitting procedures verify that the mass transfer for DMMP vapor adsorption within the SFB has been improved, and the utilization of bed capacity has also been elevated; the rate-limiting process is the diffusion of DMMP vapor to the adsorbent surface.

## Figures and Tables

**Figure 1 nanomaterials-13-02661-f001:**
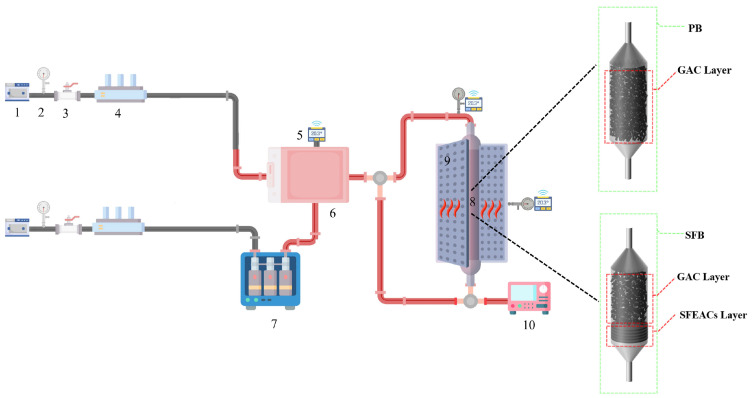
Schematic diagram of adsorption experimental setup. (1. Dryer, 2. Pressure gauge, 3. Control valve, 4. A mass flow controller, 5. Temperature sensor, 6. Preheated mixer, 7. DMMP vapor generation device, 8. Adsorption column, 9. Isothermal heating module, 10. Total hydrocarbon analyzer.).

**Figure 2 nanomaterials-13-02661-f002:**
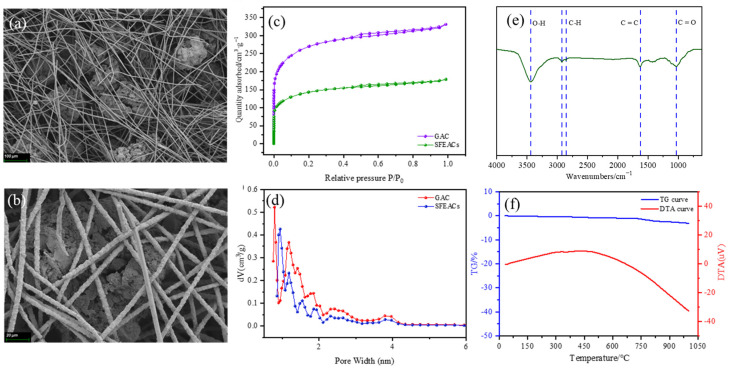
SEM images of SFEACs (**a**,**b**); N_2_ adsorption–desorption isotherms of the SFEACs and GAC (**c**). Pore size distribution of GAC and SFEACs (**d**); FTIR spectra of the SFEACs (**e**); TG and DTA curves of SFEACs (**f**).

**Figure 3 nanomaterials-13-02661-f003:**
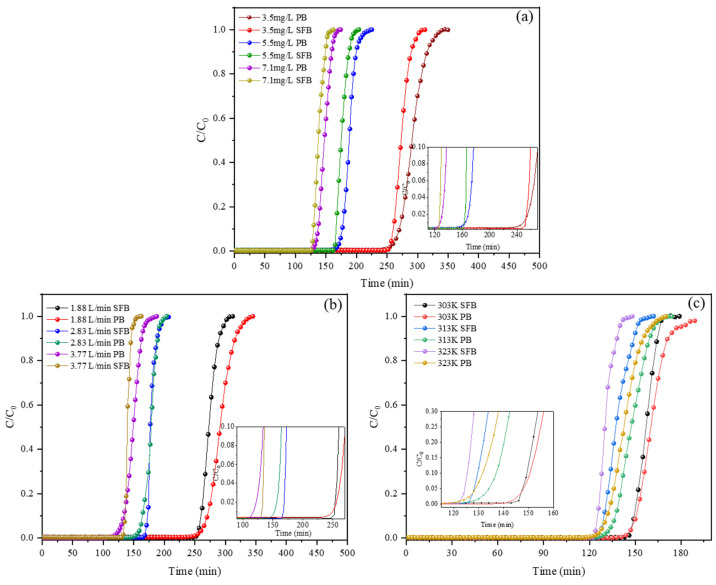
The breakthrough curves of SFB and PB on DMMP vapor at different concentrations (**a**), flow rate (**b**) and adsorption temperature (**c**).

**Figure 4 nanomaterials-13-02661-f004:**
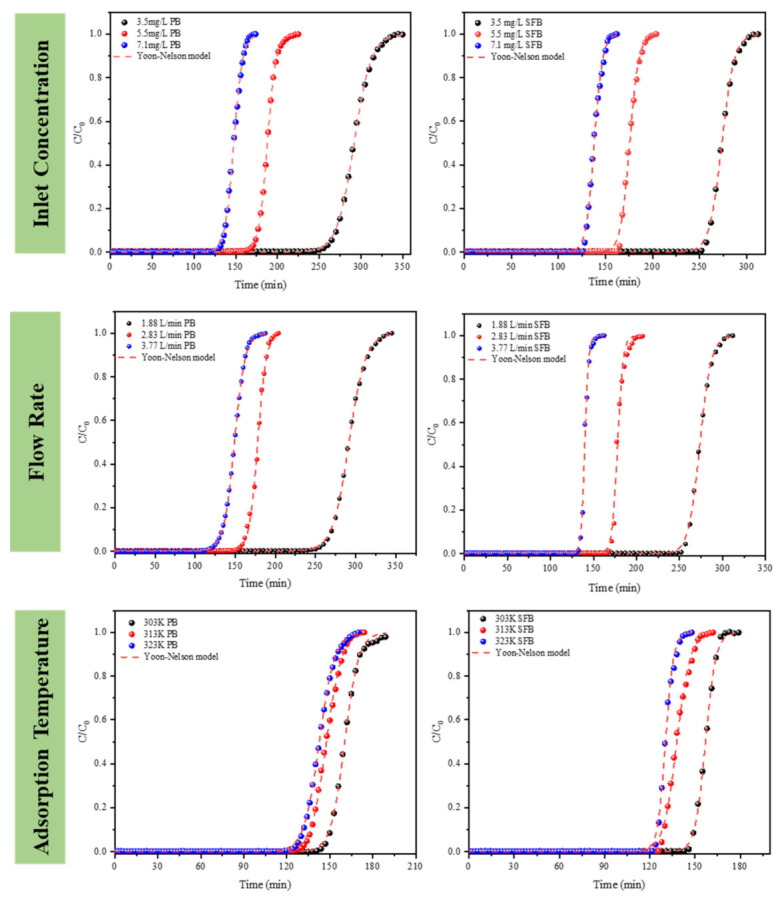
Yoon-nelson model fits DMMP vapor breakthrough curves over SFB and PB.

**Figure 5 nanomaterials-13-02661-f005:**
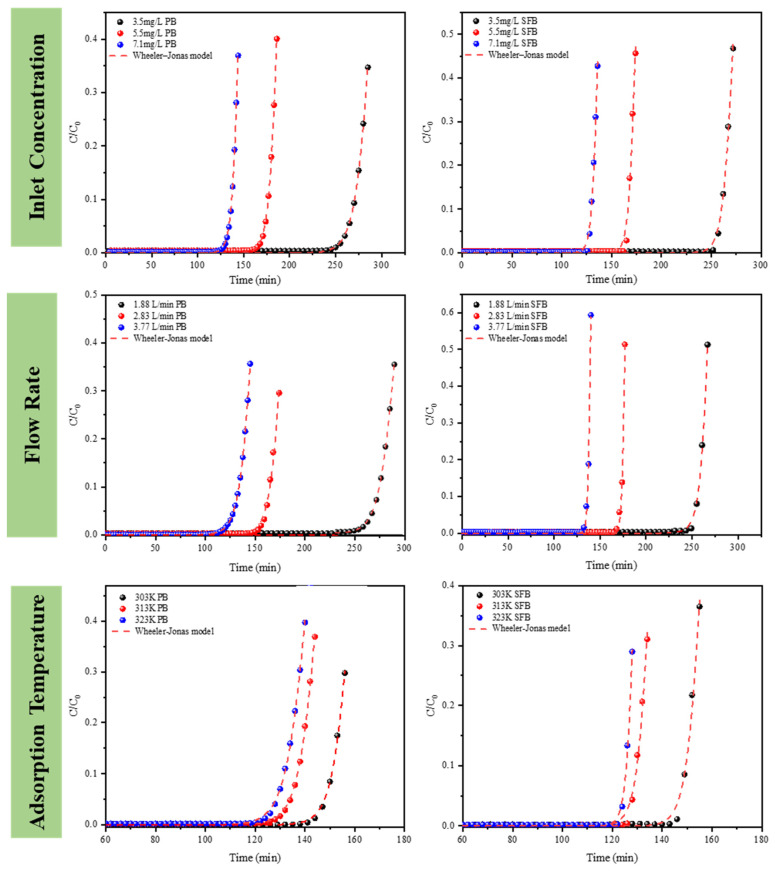
The Wheeler–Jonas model fits the initial part of the DMMP vapor breakthrough curve at SFB and PB.

**Figure 6 nanomaterials-13-02661-f006:**
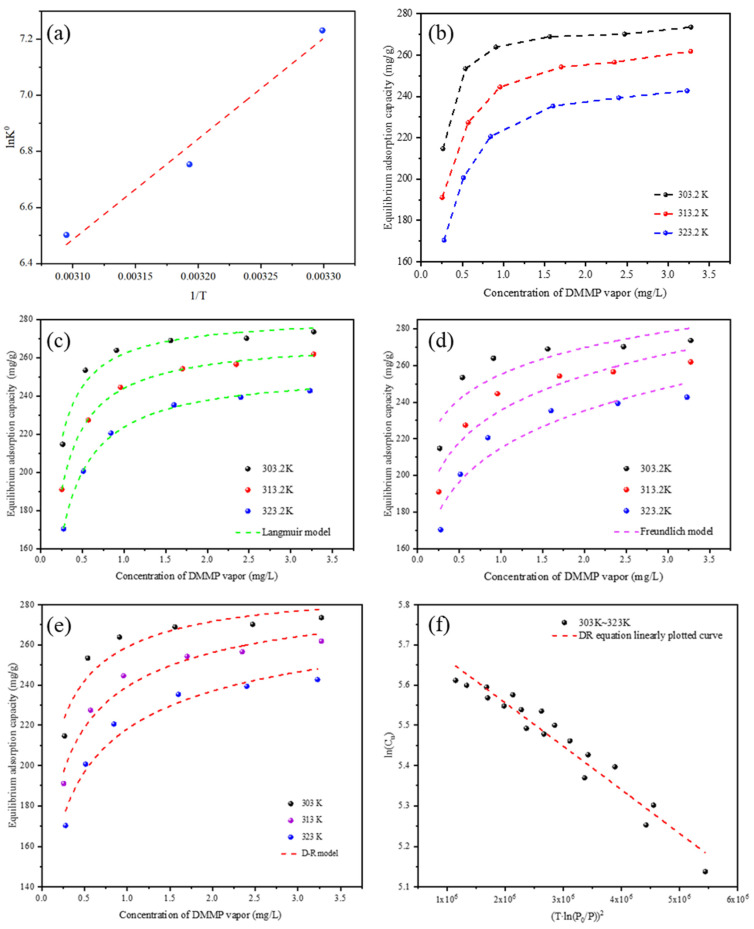
The drawing of 1/*T* − *lnK*^0^ (**a**); Adsorption isotherms of DMMP vapor onto SFEACs (**b**); Fitted using Langmuir models (**c**); Fitted using Freundlich models (**d**); Fitted using D-R models (**e**); D-R equation linear plotting curve (**f**).

**Table 1 nanomaterials-13-02661-t001:** Pore structure parameters of SFEACs and GAC before and after sintering.

Sample	S_BET_ m^2^·g^−1^	V_tot_ cm^3^·g^−1^	V_m_ cm^3^·g^−1^	V_mic_/V_tot_ %	D_Av_ nm
GAC	982	0.5126	0.358	69.84	2.09
SFEACs	520	0.2764	0.188	68.02	2.13

**Table 2 nanomaterials-13-02661-t002:** Yoon-Nelson model dynamic parameters of DMMP vapor adsorption by SFB and PB under different conditions.

Bed	Conditions			*τ_exp_* (min)	Yoon-Nelson		*R* ^2^
	*C*_0_ (mg/L)	*T* (K)	*Q* (L/min)		*τ* (min)	*k*′ (min^−1^)	
SFB	3.6	313	1.88	273.9	273.5	0.15	0.9992
	5.5			176.5	175.6	0.20	0.9979
	7.1			138.3	138.0	0.23	0.9979
PB	3.6			292.5	291.8	0.10	0.9996
	5.5			188.9	188.0	0.17	0.9998
	7.1			148.3	147.7	0.19	0.9991
SFB	3.6	313	1.88	274.7	273.5	0.15	0.9992
			2.83	179.7	178.0	0.31	0.9956
			3.77	140.5	139.9	0.46	0.9977
PB			1.88	293.0	291.8	0.10	0.9996
			2.83	192.5	193.1	0.14	0.9996
			3.77	147.6	148.9	0.15	0.9999
SFB	7.1	303.2	1.88	157.8	157.0	0.30	0.9993
		313.2		138.3	137.9	0.23	0.9979
		323.2		131.2	130.5	0.37	0.9981
PB		303.2		160.2	160.4	0.21	0.9993
		313.2		147.3	147.7	0.19	0.9991
		323.2		143.6	142.7	0.19	0.9996

**Table 3 nanomaterials-13-02661-t003:** Wheeler–Jonas model dynamic parameters of DMMP vapor adsorption by SFB and PB under different conditions.

Bed	Conditions			*W_e_* (mg/g)	*K_v_* (min^–1^)	*L_c_* (cm)	*t_b_* (min)	*R* ^2^
	*C*_0_ (mg/L)	*T* (K)	*Q* (L/min)					
SFB	3.6	313	1.88	397.4	10344.2	0.31	255	0.9979
	5.5			395.9	7098.2	0.45	164	0.9980
	7.1			401.4	6169.9	0.51	127	0.9992
PB	3.6			427.1	5550.6	0.57	249	0.9964
	5.5			423.4	6117.8	0.52	159	0.9918
	7.1			429.3	5396.5	0.59	121	0.9981
SFB	3.6	313	1.88	397.4	11027.2	0.29	255	0.9964
			2.83	397.7	15360.4	0.31	165	0.9918
			3.77	394.3	19641.3	0.34	132	0.9981
PB			1.88	427.4	5540.4	0.57	249	0.9978
			2.83	427.2	7120.0	0.67	148	0.9999
			3.77	429.3	8210.7	0.77	116	0.9964
SFB	7.1	303	1.88	456.9	8190.3	0.39	138	0.9993
		313		396.1	7558.5	0.42	127	0.9979
		323		345.2	7292.2	0.44	126	0.9980
PB		303		465.1	6550.7	0.48	136	0.9980
		313		426.7	5609.2	0.57	121	0.9971
		323		412.7	5307.2	0.60	116	0.9981

**Table 4 nanomaterials-13-02661-t004:** (**a**) Isotherm model parameters for the adsorption of DMMP vapor onto SFEACs. (**b**) Thermodynamics parameters for the adsorption of DMMP vapor onto SFEACs.

(a)				
Models	Parameters	Temperature (K)		
		303	313	323
*Langmuir*	*K_L_* (m^3^/mg)	13.25	9.56	7.47
	*q_e,cal_* (mg/g)	281.9	269.7	253.7
	*q_e,exp_* (mg/g)	285.3	270.4	255.1
	*R* ^2^	0.964	0.998	0.998
*Freundlich*	*K_F_* (L/g)	0.255	0.236	0.215
	1/*n*	0.079	0.112	0.130
	*R* ^2^	0.783	0.903	0.909
*Dubinin-Radushkevich*	*q*_0*,cal*_ (mg/g)	282.1	278.5	271.3
	*q_e,exp_* (mg/g)	285.3	270.4	255.1
	*E* (kJ/mol)	17.70	17.87	18.75
	*R* ^2^	0.904	0.965	0.958
**(b)**				
** *T* ** **(K)**	**∆*H* (kJ/mol)**	**∆*S* (J/(mol·K))**	**∆*G* (kJ/mol)**	
303.15	−23.33	−15.52	−18.63	
313.15			−18.48	
323.15			−18.32	

## Data Availability

The data that support the findings of this study are available from the corresponding author upon reasonable request.

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
