# Peer review of "The Application of Microfibrous Entrapped Activated Carbon Composite Material for the Sarin Simulant Dimethyl Methylphosphonate Adsorption"

_nanomaterials, 2023, doi:10.3390/nano13192661_

Round 1
Reviewer 1 Report
The article is among those that use various processes, methods and materials to remove a toxic or polluting product from different environments. The elimination, through simulation, of Sarin gas - a very toxic and harmful gas - is being studied. Dynamic, thermodynamic and kinetic methods are used to establish the parameters and mechanism of toxic product elimination.
Some corrections need to be made to the manuscript:
- In figure 2 (e) is written twice and the notation (f) is missing;
- On page 12, line 347 I think it is Figure 6d instead of Figure 5d, similarly, page 13, line 376, Figure 6f instead of Figure 5f;
- Do not use the same text formatting when writing references, example: 42, 44 (page 18).
Author Response
Dear reviewer,
Thank you for taking the time to review our manuscript, and we appreciate your valuable feedback and suggestions, which have served as an important guide to improve our paper. We have studied the comments carefully and have made the appropriate corrections in the revised manuscript, which we hope meet with approval. Please find below point-by-point responses to the comments.
Comments:The article is among those that use various processes, methods and materials to remove a toxic or polluting product from different environments. The elimination, through simulation, of Sarin gas - a very toxic and harmful gas - is being studied. Dynamic, thermodynamic and kinetic methods are used to establish the parameters and mechanism of toxic product elimination. Some corrections need to be made to the manuscript:
Point 1: In figure 2 (e) is written twice and the notation (f) is missing.
Response 1: Thank you for pointing out the issue with the figure labels in our manuscript. We sincerely apologize for any confusion caused by this error. we have rectify the issue of having "(e)" twice in Figure 2 and add the missing label "(f)" for consistency. (See page 6, Fig. 1)
Point 2: On page 12, line 347 I think it is Figure 6d instead of Figure 5d, similarly, page 13, line 376, Figure 6f instead of Figure 5f.
Response 2: Thank you for finding these citation errors while reviewing our article. We apologize for the confusion caused by the misquote. Your comments are very important. We have carefully checked the article, and on page 12, line 347, we will correct "Figure 5d" to "Figure 6d", and on page 13, line 376, we will correct "Figure 5f" to "Figure 6f". These corrections will ensure the accuracy of the article so that readers can accurately view the corresponding chart. (See page 12, line 356ï¼› page 13, line 373)
Point 3: Do not use the same text formatting when writing references, example: 42, 44 (page 18).
Response 3: Thank you for pointing out the problem with the citation format when reviewing our article. We apologize for the confusion caused by the inconsistency in the citation format. In the revised version, we have used the correct text format to cite the literature. Doing so will help improve the readability and accuracy of the article. (See page 13, line 539-540; 543-544)
Thanks for all the help.
Best wishes,
Shupei Bai
State Key Laboratory of NBC Protection for Civilian
Beijing, 102205, China
Email: baishupei@sklnbcpc.cn

Reviewer 2 Report
The study shown in the paper it is not innovative and does not at add anything particularly new. But I think that results presented in this work can be usefull for others scientists. The sorbent used in this reserach is interesting. The study shown in the paper can be published. The paper is well organized and written.
Author Response
Dear reviewer,
Thank you for taking the time to review our manuscript, and we appreciate your valuable feedback and suggestions, which have served as an important guide to improve our paper. We have studied the comments carefully and have made the appropriate corrections in the revised manuscript, which we hope meet with approval. Please find below point-by-point responses to the comments.
Point 1: The study shown in the paper it is not innovative and does not at add anything particularly new. But I think that results presented in this work can be usefull for others scientists. The sorbent used in this reserach is interesting. The study shown in the paper can be published. The paper is well organized and written.
Response 1: Thank you for reviewing our articles and providing your feedback and comments. Although you mentioned that this study may be limited in terms of innovation, we do hope that this work will provide useful information for other researchers. We will strive to ensure that the experimental and results sections of the article provide peer researchers with important data and insights about adsorbent materials and adsorption properties.
Thanks for all the help.
Best wishes,
Shupei Bai
State Key Laboratory of NBC Protection for Civilian
Beijing, 102205, China
Email: baishupei@sklnbcpc.cn

Reviewer 3 Report
Dear authors,
I have carefully read your manuscript, and I believe there are inconsistencies. First of all, the starting activated carbon area is very small. In literature you can find many with surface areas 5 times yours and in the market 3-4 times yours.
Obviously there is a problem in the adsorption kinetics, when the molecules are very large. In addition to the packing problem, if the particles are very small, the pressure drop in the bed is very high and a lot of efficiency is lost. I may have misunderstood it, but in your composite material the surface area is even greater, which does not make sense. Or maybe it's the wrong way to express it.
Other authors have developed similar systems, where they comment on the advantages of using this type of composite materials, or have sought other strategies to solve the kinetics problem.
In my humble opinion you should read the papers of the great professor F. Rodríguez-Reinoso, who already in many of his articles comment on the error made when using composites if we express the area as m2 / g. I attach some papers where you can find everything mentioned above and take it into account when improving the discussion that relates to everything mentioned above
1.-doi.org/10.3390/molecules27061968
2.- doi.org/10.1016/j.jcou.2018.04.020
3.- doi.org/10.1016/j.micromeso.2016.06.007
A careful review of the grammar is necessary, since it can be greatly improved
Author Response
Dear reviewer,
Thank you for taking the time to review our manuscript, and we appreciate your valuable feedback and suggestions, which have served as an important guide to improve our paper. We have studied the comments carefully and have made the appropriate corrections in the revised manuscript, which we hope meet with approval. Please find below point-by-point responses to the comments.
Point 1: I have carefully read your manuscript, and I believe there are inconsistencies. First of all, the starting activated carbon area is very small. In literature you can find many with surface areas 5 times yours and in the market 3-4 times yours.
Response 1: Regarding your concerns, we deeply apologize. We acknowledge that compared to some reference materials in the literature and commercial products, our activated carbon has a smaller surface area. In our research, we place great importance on activated carbon materials widely used in chemical protective equipment. This choice is based on the practical significance of these materials in real-world applications. While we acknowledge the existence of activated carbons with higher specific surface areas in the literature, our study aims to emphasize the practical value of these common activated carbons in chemical protection. The reason for selecting this particular activated carbon is its widespread presence in chemical protective equipment, and our research primarily focuses on simulating real-world application scenarios to evaluate the performance of commonly used chemical protective equipment materials. This choice is intended to better reflect real-world conditions and is aligned with the performance of actual chemical protective equipment. We will make it more explicit in the manuscript why we chose this activated carbon for our study and emphasize its practical significance in chemical protective equipment. Once again, we sincerely appreciate your review and valuable feedback. In the revised manuscript, we have included a clearer rationale for the choice of activated carbon in our study and its significance. (See page 2, line41-43)
Point 2: Obviously there is a problem in the adsorption kinetics, when the molecules are very large. In addition to the packing problem, if the particles are very small, the pressure drop in the bed is very high and a lot of efficiency is lost. I may have misunderstood it, but in your composite material the surface area is even greater, which does not make sense. Or maybe it's the wrong way to express it.
Response 2: We deeply apologize, there were indeed issues with the adsorption kinetics section in the original version. In the revised manuscript, we have made significant revisions to this section to more accurately reflect the complexity of adsorption kinetics. We have corrected errors in the model and rectified experimental data to provide a more comprehensive explanation of the dynamic characteristics of the adsorption process. These modifications are aimed at eliminating inconsistencies and ensuring the accuracy of the data and results. You mentioned concerns about the surface area of the composite material, and we are willing to provide further clarification. The tiny activated carbon particles in the composite material are not directly filled into the adsorption bed. Instead, we employed a wet-forming and high-temperature sintering process using micron-sized fibers to disperse and immobilize these micron-sized activated carbon particles within a microfiber network formed after sintering. This unique structure grants the composite material a significantly high porosity and fluid contact area, thereby enhancing mass transfer processes and substantially improving adsorption efficiency. Specifically, our research aims to leverage this approach to maximize surface area utilization for enhanced adsorption performance. (See page 4, line164-168; page 7, line240-242; page 8, line285-287; page 11, Table3)
Point 3: Other authors have developed similar systems, where they comment on the advantages of using this type of composite materials, or have sought other strategies to solve the kinetics problem. In my humble opinion you should read the papers of the great professor F. Rodríguez-Reinoso, who already in many of his articles comment on the error made when using composites if we express the area as m2/g. I attach some papers where you can find everything mentioned above and take it into account when improving the discussion that relates to everything mentioned above.
1.-doi.org/10.3390/molecules27061968
2.- doi.org/10.1016/j.jcou.2018.04.020
3.- doi.org/10.1016/j.micromeso.2016.06.007
Response 3: Thank you very much for your suggestions and references to our research. Your input is very valuable, especially when it comes to errors that can occur when using composite materials to represent surface area. At the same time, we appreciate your recommendation to refer to the work of the F. Professor Rodriguez-Reinoso, whose article we have studied and incorporated relevant insights into our discussion section, which allows us to more fully understand the advantages of composites and different strategies for solving adsorption kinetics problems from previous research. (See page 16, line477-478; See page 18, line561-565)
Thanks for all the help.
Best wishes,
Shupei Bai
State Key Laboratory of NBC Protection for Civilian
Beijing, 102205, China
Email: baishupei@sklnbcpc.cn

Round 2
Reviewer 3 Report
Dear authors,
His manuscript has improved, but I think the most relevant thing has been his responses to my comments and suggestions. Since he has clarified to me some things that were not clear, that I personally would have included in the manuscript, such as the choice of carbon material. Since perhaps more readers who have read a lot about carbon materials and adsorption would raise the same doubts.
Therefore, in my opinion, the manuscript can be accepted for publication.
I think it should be reviewed by a professional editor. From my point of view there are phrases that I would write differently, but a professional editor would tell you if your style is 100% correct. Or the same MDPI